# Quantitative Measurement of Progesterone Receptor Immunohistochemical Expression to Predict Lymph Node Metastasis in Endometrial Cancer

**DOI:** 10.3390/diagnostics12040790

**Published:** 2022-03-23

**Authors:** Yu-Yang Hsiao, Hung-Chun Fu, Chen-Hsuan Wu, Jui Lan, Yu-Che Ou, Ching-Chou Tsai, Hao Lin

**Affiliations:** 1Department of Obstetrics and Gynecology, Kaohsiung Chang Gung Memorial Hospital and Chang Gung University College of Medicine, Kaohsiung 83341, Taiwan; stribte@cgmh.org.tw (Y.-Y.H.); allen133@cgmh.org.tw (H.-C.F.); chenhsuan@adm.cgmh.org.tw (C.-H.W.); ou4727@cgmh.org.tw (Y.-C.O.); nick@adm.cgmh.org.tw (C.-C.T.); 2Department of Obstetrics and Gynecology, Chia-Yi Chang Gung Memorial Hospital, Chia-Yi 61363, Taiwan; 3Department of Anatomic Pathology, Kaohsiung Chang Gung Memorial Hospital and Chang Gung University College of Medicine, Kaohsiung 83341, Taiwan; blueray@cgmh.org.tw

**Keywords:** progesterone receptor immunohistochemical staining, grade, CA125, endometrial cancer, lymph node metastasis

## Abstract

Background: Previous studies have shown that loss of progesterone receptor (PR) in endometrial cancer (EC) is associated with poor outcomes. Evaluating lymph node metastasis (LNM) is essential, especially before surgical staging. The aim of this study was to investigate the role of PR expression and other clinicopathological parameters in LNM and to develop a prediction model. Methods: We retrospectively evaluated endometrioid-type EC patients treated with staging surgery between January 2015 and March 2020. We analyzed PR status using immunohistochemical staining, and the expression was quantified using the H-score. We identified optimal cut-off values of H-score and CA125 for predicting LNM using receiver operating characteristic curves, and used stepwise multivariate logistic regression analysis to identify independent predictors. A nomogram for predicting LNM was constructed and validated using bootstrap resampling. Results: Of the 310 patients evaluated, the optimal cut-off values of PR H-score and CA125 were 162.5 (AUC 0.670, *p* = 0.001) and 40 U/mL (AUC 0.739, *p* < 0.001), respectively. Multivariate analysis showed that CA125 ≥ 40 U/mL (OR: 8.03; 95% CI: 3.44–18.77), PR H-score < 162.5 (OR: 5.22; 95% CI: 1.87–14.60), and tumor grade 2/3 (OR: 3.25; 95% CI: 1.33–7.91) were independent predictors. These three variables were incorporated into a nomogram, which showed effective discrimination with a concordance index of 0.829. Calibration curves for the probability of LNM showed optimal agreement between the probability as predicted by the nomogram and the actual probability. Our model gave a negative predictive value and a negative likelihood ratio of 98.4% and 0.14, respectively. Conclusions: PR H-score along with tumor grade and CA125 are helpful to predict LNM. In addition, our nomogram can aid in decision making with regard to lymphadenectomy in endometrioid-type EC.

## 1. Introduction

Endometrial cancer is one of the most common malignant diseases in women, and according to cancer registry annual reports, it is the most common gynecologic cancer in Taiwan. There were 2439 newly diagnosed cases of endometrial cancer in Taiwan in 2018, and nearly 70% of these patients were diagnosed with International Federation of Gynecology and Obstetrics (FIGO) stage I disease [1,2]. Surgical staging with total hysterectomy, bilateral salpingo-oophorectomy, and pelvic/para-aortic lymphadenectomy remains the standard treatment for patients with endometrial cancer. Lymph node metastasis (LNM) is a significant prognostic factor, causing a higher recurrence rate and lower survival rate. However, the need for routine pelvic/para-aortic lymphadenectomy is still a matter of debate [3,4,5,6,7,8,9], especially for patients with presumed clinical stage I disease, which has a better survival rate and lower incidence of LNM. This debate is due in part to the results of two randomized controlled trials which demonstrated that systematic lymphadenectomy had no survival benefit in patients with early-stage endometrial cancer compared to those who did not undergo lymphadenectomy [3,4]. Routine lymphadenectomy can provide precise disease staging and guide postoperative adjuvant therapy. However, it can also lead to increased operating time, blood loss, and other postoperative morbidities including lymphedema, lymphocele, and lymphatic/chylous ascites, which require further intervention [10]. Therefore, gynecologic oncologists who do not perform routine lymphadenectomy choose the adjuvant therapy based on age and uterine risk factors such as depth of myometrial invasion and lymphovascular invasion [11,12]. Several studies have investigated which biomarkers may help to identify patients at risk of LNM, including tumor markers such as cancer antigen 125 (CA125), imaging studies, and pathologic parameters [13,14,15]. Our team was the first to identify that a pretreatment CA125 level >40 U/mL could be a criterion for systematic lymphadenectomy with satisfactory sensitivity and specificity [16]. Since then, several studies have used CA125 as a possible indicator for LNM, most of which have reported similar results but with slightly different cut-off values.

Endometrioid-type endometrial cancer is a hormonally regulated disease, and positive progesterone receptor (PR) status has been associated with a favorable prognosis [17]. Moreover, progestin therapy can be an option for selected patients, such as young women who wish to preserve fertility. In contrast, double-negative hormone receptor or negative PR status has been associated with shorter progression-free survival (PFS) and overall survival (OS) [18,19]. However, only a few studies have discussed the role of PR status in estimating the risk of LNM, and therefore, the predictive value of PR status remains unclear. The aim of this study was to evaluate whether integrating the expression of PR and other clinicopathological parameters into clinical risk stratification can help to predict LNM in patients with endometrial cancer.

## 2. Materials and Methods

### 2.1. Study Population

In this retrospective cohort study, we screened 558 patients with endometrial cancer who received treatment at Kaohsiung Chang Gung Memorial Hospital between January 2015 and March 2020. Clinical data were reviewed from medical records. This study was approved by the Institutional Review Board (IRB) of Chang Gung Memorial Hospital. The requirement of informed written consent was waived by the IRB.

The inclusion criteria were patients with endometrioid-type endometrial cancer, and those with available data on pretreatment CA125 levels, status of hormone receptors, basic characteristics, and clinicopathological parameters. In addition, only patients who received comprehensive staging surgery including total hysterectomy, bilateral salpingo-oophorectomy, pelvic lymphadenectomy, and collection of peritoneal washing fluid for cytology examinations were included. Patients were included regardless of whether or not they received para-aortic lymphadenectomy in the staging surgery, and those who received laparotomy or minimally invasive surgeries were also included. We excluded patients with non-endometrioid histology, those without comprehensive staging surgery, and those whose background characteristics were not available.

### 2.2. CA125, Grading, and PR Immunohistochemical Staining Measurement

The Architect CA125 II assay, a chemiluminescent microparticle immunoassay (CMIA), was used for the quantitative determination of serum CA125. Serum samples were analyzed before the staging surgery, and the results were retrieved from clinical records. Tumor grade and immunohistochemical (IHC) staining of PR were evaluated in the endometrial tumors after the hysterectomy. The grading system was based on the percentage of solid tumor growth, according to the FIGO criteria [2]. Formalin-fixed, paraffin-embedded tissue sections were obtained to evaluate the expressions of PR. Sections were deparaffinized with xylene, rehydrated with a graded alcohol series (100%, 95%, 85%, and 75%), and then rinsed with distilled water. Antigen retrieval was enhanced using citrate buffer (10 mM, pH 6.0). Endogenous peroxidase activity was quenched by incubation in 3% hydrogen peroxide solution. The slides were incubated with a primary antibody against PR (Leica, US. Cat# PR NCL-L-PGR-312), which can catch both PR isoforms (PR-A and PR-B), and then further incubated with a secondary antibody. Antigen–antibody complexes were detected using diaminobenzidine (DAB) (Dako, Glostrup, Denmark) and counterstained with Gill’s hematoxylin (Merck, Whitehouse, NJ, USA). Slides were visualized at 200× magnification, and PR staining was scored using the H-score method (range 0–300) obtained by multiplying the tumor nuclei cell intensity (on a scale of 0 to 3) by the percentage of positive tumor nuclei cells (on a scale of 0 to 100) [20]. One pathologist (Lan J) blindly scored all cases.

### 2.3. Statistical Analysis

We attempted to identify optimal cut-off values of CA125 and PR H-score by determining the point with maximal Youden index (sensitivity + specificity −1) in receiver operating characteristic (ROC) curve analysis. After using univariate and multivariate logistic regression models, a prediction model was developed. We calculated the odds ratios (ORs) and corresponding 95% confidence intervals (CIs) to reflect the impact of different variables. To compare the predictive performance in different prediction models, we estimated positive/negative predictive values, the likelihood ratio ([1 − sensitivity]/specificity), and the post-test probability of our prediction model. The discrimination performance of this model was determined by calculating the area under the ROC curve (AUC). The calibration of the model was determined using the Hosmer–Lemeshow goodness-of-fit test.

Based on our multivariate prediction model, we developed a nomogram to calculate the risk of LNM. Each risk factor had different points, and a higher point indicated a higher risk of LNM. The discrimination was assessed with 300 bootstrap resamples. We calculated the concordance index (c-index) for each bootstrap sample, which represents the model’s predictive accuracy. Calibration plots were assessed, which showed how far the predicted probabilities were from the actual observed proportion with LNM. Data management and analysis were performed using SPSS software for Windows version 22 (IBM, Armonk, NY, USA). A *p*-value less than 0.05 was considered to be statistically significant. The statistical analysis of nomograms was performed using R 3.1.1 software, available online.

## 3. Results

### 3.1. Patient Characteristics

After reviewing the medical records, 310 patients who met the inclusion criteria were finally enrolled for analysis (Figure 1). The detailed characteristics of the study population are shown in Table 1. The mean age at diagnosis was 55 years. About one-fourth of the patients were nulliparous. A majority of the patients were diagnosed at stage I disease, of whom 64.8% had FIGO stage IA and 15.5% had stage IB. Almost half of the tumors had grade 1 histology. The rates of more than half myometrial invasion and the presence of lymphovascular space invasion were 28.7% and 28.1%, respectively. All of the patients received pelvic lymph node dissection, and nearly 80% also underwent para-aortic lymphadenectomy. The median numbers of retrieved lymph nodes in pelvic and para-aortic areas were 31 and five, respectively. LNM was observed in 33 (10.6%) patients. Only one case had isolated para-aortic LNM. The median PR H-score in this study cohort was 120, with a range of 0–300. Representative cases of quantitative measurement of PR IHC expression are shown in Figure 2.

### 3.2. Optimal Cut-Off Values of CA125 and PR H-Score

To determine the optimal cut-off values of CA125 level and PR H-score, ROC curves were drawn with LNM as the endpoint. The AUCs of CA125 and PR H-score were 0.739 and 0.670, respectively (*p* < 0.001). The best cut-off values of CA125 and PR H-score were 40 and 162.5, respectively, after calculating the Youden index.

### 3.3. Impact of Predictive Variables on Lymph Node Metastasis

We included six variables which could be assessed before the staging surgery into our analysis: age at diagnosis, body mass index (BMI), parity, pretreatment CA125 level, tumor grade, and PR H-score. In univariate analysis, CA125 ≥ 40 U/mL, PR H-score < 162.5, and non-grade 1 tumor were correlated with LNM (Table 2). In multivariate stepwise logistic regression analysis, these three variables (CA125, OR 8.03, 95% CI 3.44–18.77; PR H-score, OR 5.22, 95% CI 1.87–14.60; non-grade 1 tumor, OR 3.25, 95% CI 1.33–7.91) remained significant independent risk factors for LNM (Table 2). Figure 3 shows a comparison of AUCs between the variables alone and the prediction model. When we combined these three independent risk factors, the AUC of our model was 0.818 (95% CI, 0.771–0.859). The Hosmer–Lemeshow goodness-of-fit test showed high stability of this logistic predictive model (*p* = 0.913).

### 3.4. Nomogram Construction and Performance of Prediction Model

A nomogram predicting the probability of LNM in the patients with endometrioid endometrial cancer is shown in Figure 4A. The calibration plot for the prediction model is shown in Figure 4B. The mean absolute error was 0.011 and the C-index was 0.829, both indicating adequate performance. We defined the low-risk group as CA125 <40 U/mL, PR H-score >162.5, and grade 1 tumor, which had the lowest risk of LNM (<1%). A total of 60 patients (19.4%) were classified as being at low risk. Among these patients, one (false negative rate = 1.6%) had LNM, and the negative predictive value was 98.3% (95% CI, 0.91–0.99). The sensitivity and specificity of the model were 97.0% and 21.7%, respectively. The negative likelihood ratio of the model was 0.14 (95% CI, 0.02–0.97), with moderate evidence to rule out the possibility of LNM. If the prevalence of LNM in endometrial cancer was set as 10%, the negative post-test probability was 1.5%. This study was not intended to rule in patients with a high risk of LNM. However, interestingly, when the patients had all three risk factors, the prediction model showed suitable specificity and high positive predictive value for predicting LNM. The positive likelihood ratio was 8.92 (95% CI, 4.99–15.92) and the positive post-test probability was 49.7% (95% CI, 35.6–63.9%). Detailed results of the predictive performance for the different variables and different combinations are shown in Table 3.

The patient who was falsely classified as being at low risk was a 48-year-old female with no history of systemic diseases. She had been diagnosed with grade 2 endometrioid endometrial cancer at another hospital. Her pretreatment CA125 level was 22 U/mL. Preoperative magnetic resonance imaging (MRI) revealed no metastatic lesion, and the clinical stage was IA. She received robotic-assisted laparoscopic staging surgery. The final pathology revealed a 1.5 cm grade 1 tumor with less than half myometrial invasion and absence of lymphovascular space invasion (LVSI), but with one right pelvic LNM. IHC staining showed a high expression of PR with an H-score of 285. Adjuvant chemotherapy was suggested, but she refused. She received regular follow-up at our outpatient department and no relapse was found during a follow-up of 37 months.

## 4. Discussion

To the best of our knowledge, this is the first study to demonstrate that combining quantitative measurement of PR IHC staining with tumor grade and pretreatment serum CA125 level can be used to predict the risk of LNM in patients with endometrioid endometrial cancer. Our prediction model had a low false negative rate and a satisfactory negative predictive value in the low-risk group, suggesting that lymphadenectomy can be omitted in low-risk patients.

Whether or not to perform lymphadenectomy is still a controversial issue. Many studies have evaluated the diagnostic and therapeutic value of pelvic and/or para-aortic lymphadenectomy. Some studies have reported that lymphadenectomy had a diagnostic role but not a therapeutic role, with no benefit on disease-free survival or OS, especially for young patients and early-stage disease [3,4,5,6,7]. However, other studies have reported that lymphadenectomy may improve recurrence-free survival (RFS) and OS if as many pelvic lymph nodes as possible are removed. In a study on stage IB-IIIC2 patients, Kim et al. reported that removing ≤14 pelvic nodes was associated with poor RFS and OS [8]. In addition, Papathemelis et al. suggested that removing ≥25 pelvic and paraaortic lymph nodes in high-grade patients (type I grade 3, type II endometrial cancer, and carcinosarcoma) could reduce the recurrence rate and improve long-term RFS and OS [9]. To provide a stronger recommendation, two ongoing randomized phase III trials (JCOG1412, ECLAT) are evaluating the actual therapeutic role of pelvic and/or paraaortic lymphadenectomy in endometrial cancer [21].

Since lymphadenectomy may be omitted in low-risk patients according to the previous studies, many gynecologic oncologists have attempted to create a prediction model to identify those at low risk. Our previous study showed that elevated CA125 was significantly correlated with many clinicopathological factors (but not correlated with tumor grade) and an increased risk of LNM (OR 8.7 for a CA125 cut-off value of 40 U/mL) [16]. Several subsequent studies combined CA125 and other parameters to develop prediction models for LNM. Kang et al. used CA125 (with a cut-off value of 35 U/mL) and three MRI parameters (deep myometrial invasion, lymph node enlargement >1 cm, and extension beyond the uterine corpus) to identify patients at low risk of LNM. The low-risk group was characterized by the absence of these four parameters, and 53% of their study population was classified as being at low risk. Their model showed an adequate predictive performance with an AUC 0.85 and negative likelihood ratio of 0.11 [13]. Later, Kang et al. conducted a prospective validation study at 20 hospitals in three Asian countries and confirmed that their low-risk criteria were reliable and accurate [22]. Todo et al. proposed a preoperative scoring system and classified the risk of LNM based on three preoperative parameters: serum CA125, tumor volume index measured by MRI, and histological grade of an endometrial biopsy. Patients who had an “LNM score 0” (low CA125, defined as <70 U/mL if aged <50 years or <28 U/mL if aged ≥50 years, non-grade 3 tumor, and low volume index defined as <25) had a 3.6% risk of pelvic LNM [23]. Mitamura et al. combined LNM score 0 with less than half myometrial invasion intraoperatively, and found that lymphadenectomy could be safely omitted in these patients [14]. Kazuaki et al. used the same clinical parameters as the LNM score with a slightly different cut-off value (only grade 1 tumors were classified as being low risk) to estimate the risk of LNM, and demonstrated similar results [24]. Therefore, combining CA125 with other preoperative risk factors appears to be useful in predicting the risk of LNM.

Type I endometrial cancer is an estrogen-dependent disease, and the pathogenesis is thought to be related to prolonged unopposed estrogen stimulation of the endometrium, with progesterone having the opposite reaction. Loss of PR may increase proliferation, which may lead to carcinogenesis and tumor progression [25]. Several studies have evaluated the relationship between the expression of hormone receptors and the prognosis of endometrial cancer. Zhang et al. conducted a meta-analysis to compare the OS, cancer-specific survival, and PFS between higher and lower levels of estrogen receptor (ER) and PR. Although publication bias and heterogeneity were found between the studies, the results still showed a significant trend that higher levels of ER and PR could predict favorable survival [17]. Smith et al. used the Allred score and classified patients into a low, medium, and high expressions of ER and PR with different cut-off values [26]. A higher hazard ratio of survival was found with lower expressions of ER and PR, and lower ER and PR expressions were correlated with higher FIGO stage, higher tumor grade, non-endometrioid histology, presence of LVSI, and higher BMI. Since the expressions of hormone receptors can be obtained from a preoperative endometrial biopsy, some studies have evaluated the possibility of using hormone receptors to predict the risk of LNM. Marcos et al. combined the IHC expressions of ER and PR with LVSI to calculate the risk of LNM in those with a low or intermediate risk (FIGO IA grade 1–3 and IB grade 1/2) [27]. Although the study population was small, their results showed an increased risk of LNM in those with an ER expression <30% or PR expression <15%. Casper et al. developed a Bayesian network named ENDORISK (preoperative risk stratification in endometrial cancer) with nine parameters, including CA125, preoperative tumor grade, and expression of hormone receptor [28]. In external validation using a prospective study cohort, the AUC for LNM was 0.82, 55.8% of the patients were classified as having a <5% risk of LNM, and the false negative rate was 1.6%. A recent study also used serum CA125 and the expression of PR to develop a prediction model [15]. The low-risk group was defined as those with serum CA125 <30 U/mL, and either or both positive PR staining >50% and Ki 67 <40%. In a total of 370 patients, 229 (61.9%) were classified as being at low risk. This model showed suitable predictive performance with an AUC of 0.82 and a negative predictive value of 97.4%. The negative likelihood ratio and negative post-test probability were 0.23 and 2%, respectively. However, how low a predictive probability of LNM can be considered to be a negligible risk is still under debate. In 1997, Boronow et al. defined <4% as a negligible risk [29], and several subsequent prediction models used this criterion for risk stratification [13,15]. If we defined <4% as a low risk in our study, either tumor grade 2/3 or lower PR H-score would be included in the low-risk group, which had a 2–3% risk of LNM according to the nomogram. A total of 159 patients (51.3%) would have been classified into this subgroup, of whom four (2.5%) had LNM. This predictive performance is still comparable to the aforementioned studies. Furthermore, our nomogram could estimate the risk of LNM with different combinations of predictive parameters, and it could help when making treatment decisions and counseling the patients.

There are several strengths to this study. First, some risk scoring systems using MRI parameters to identify patients at low risk of LNM have been developed with satisfactory predictive performance. However, MRI is expensive and may not be available in low-resource areas. In addition, some patients undergo non-enhanced MRI due to comorbidities, and this would influence the accuracy of the imaging study. Currently, most cases of endometrial cancer can be diagnosed based on an endometrial biopsy through pipelle sampling or curettage. Even a small amount of endometrial tissue can provide information on the histologic type, tumor grade, and IHC staining. These preoperative predictive factors are applicable to all patients and are cost-effective. Second, some patients are incidentally diagnosed as having endometrial cancer after simple hysterectomy. Our predictive model can provide the probability of LNM and be used as reference for further re-staging consideration. Third, we used both the proportion of IHC staining to represent the expression of PR and also the intensity of nuclear staining. In breast cancer, the American Society of Clinical Oncology/College of American Pathologists guidelines recommend reporting hormone receptor test results in a semi-quantitative manner [30]. There are currently two commonly used scoring methods: the Allred score and H-score. Unlike the Allred score, which needs to translate the proportion of IHC staining to a score from 0 to 5, the H-score can be simply calculated by multiplying the proportion and intensity of the staining, to provide a total score ranging from 0 to 300. The H-score has the benefits of a wide dynamic range and can better represent the degree of hormone receptor expression. Fourth, nearly 80% of the patients received para-aortic lymphadenectomy in our study group, which is relatively higher than in previous studies. Therefore, we could report a more precise status of LNM.

There are also some limitations to the current study. First, PR has two major isoforms, A and B, and a previous study had reported that only PR-B was associated with a better prognosis [31]. In present study, we measured total PR expression, and we did not know which isoform was predominant. Second, this was a retrospective study from a single center, and the extent of lymphadenectomy was inconsistent within the study population, which may have underestimated the risk of LNM. Third, the results of tumor grade and PR IHC staining in our study were collected from hysterectomy specimens. Previous studies have suggested a moderate correlation between tumor grade in a preoperative biopsy and the final pathology, with an accuracy ranging from 56% to 89% [14,32,33]. Discordance, especially in a greater tumor size, may have influenced the interpretation of the true predictive probability [31]. This may also have occurred in PR IHC staining. Therefore, combining other risk factors into the prediction model would be a reasonable and practical method to improve the accuracy of the prediction performance. In the future, a large-scale, multi-center study design to validate the utility of our model with inclusion of different PR isoforms measurement is needed.

## 5. Conclusions

In conclusion, our results demonstrated that a higher PR expression (defined as H-score ≥162.5), tumor grade 1, and lower pretreatment CA125 level (defined as <40 U/mL) could be used to identify patients at low risk of LNM, and our nomogram can aid in decision making regarding lymphadenectomy in patients with endometrioid-type endometrial cancer.

## Figures and Tables

**Figure 1 diagnostics-12-00790-f001:**
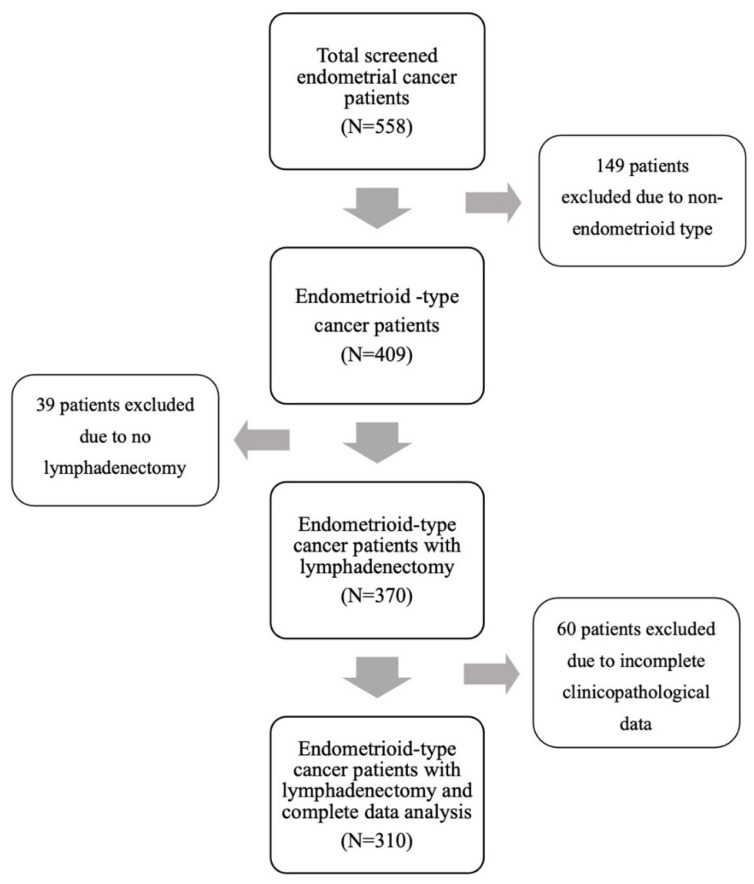
CONSORT flow diagram of study population.

**Figure 2 diagnostics-12-00790-f002:**
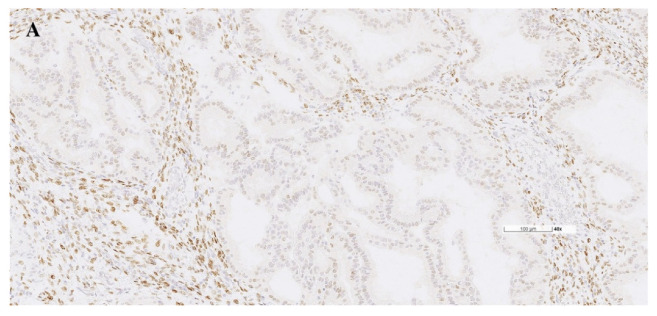
Representative cases of quantitative measurement of immunohistochemical expression of PR in endometrioid-type endometrial cancer. (**A**) Weak expression in 30% of the tumor cells with an H-score of 1 × 30 = 30; (**B**) Strong expression in 95% of the tumor cells with an H-score of 3 × 95 = 285.

**Figure 3 diagnostics-12-00790-f003:**
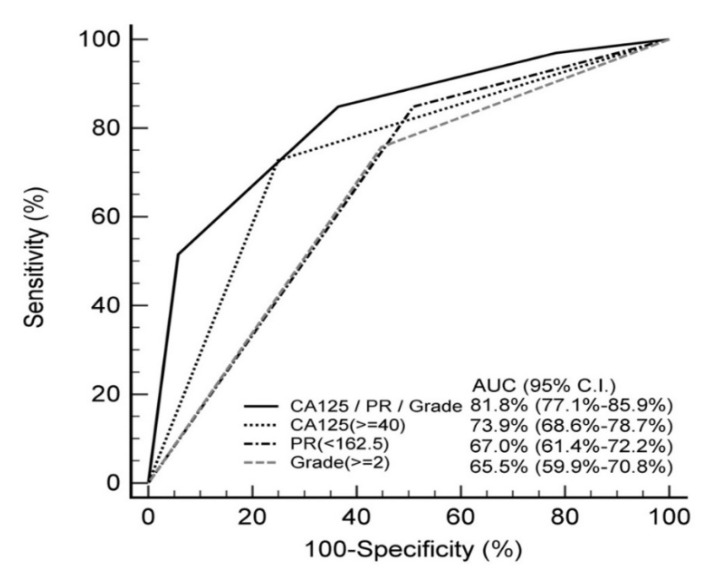
Comparison of area under ROC curve in different variables.

**Figure 4 diagnostics-12-00790-f004:**
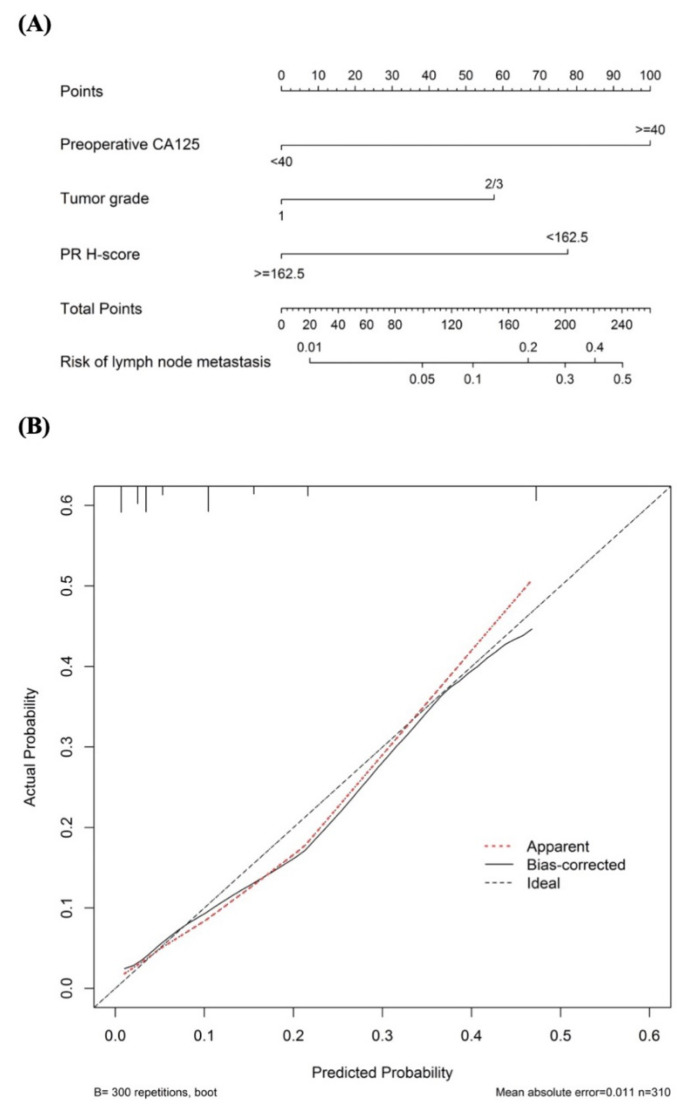
(**A**) A nomogram predicting the probability of lymph node metastasis in patient with endometrioid endometrial cancer. The probability is calculated by drawing a line to the point on the axis for each of the following variables: preoperative CA125, tumor grade, and PR H-score. The points for each variable are summed and located on the total point line. Next, a vertical line is projected from the total point line to the predicted probability bottom scale to obtain the individual probability of lymph node metastasis; (**B**) The calibration plot for the prediction model.

**Table 1 diagnostics-12-00790-t001:** Clinicopathological characteristics of the study population.

Variable	N = 310	%
Age (years)		
Median, range	55, 24–83	
Parity		
No	79	25.5
Yes (≥1)	231	74.5
Body mass index (kg/m^2^)		
Median, range	25.5, 15.1–54.1	
FIGO stage		
IA	201	64.8
IB	48	15.5
II	13	4.3
III	38	12.2
IV	10	3.2
Pathologic tumor size		
≤2cm	89	28.7
>2cm	221	71.3
LVSI		
No	223	71.9
Yes	87	28.1
Myometrial infiltration		
<1/2	221	71.3
≥1/2	89	28.7
Grade		
1	161	51.9
2	117	37.8
3	32	10.3
Type of lymphadenectomy		
Pelvic only	63	20.3
Pelvic and para-aortic	247	79.7
No. of harvested LNs		
Pelvic lymph node		
Median, range	31, 1–105	
Para-aortic lymph node		
Median, range	5, 1–47	
Lymph node metastasis		
No	277	89.4
Yes	33	10.6
Para-aortic involvement	14	42.4 (14/33)
Para-aortic only	1	3.0 (1/33)
CA125 (U/mL)		
Median, range	22.4, 2.1–5701.7	
PR (H-score)		
Median, range	120, 0–300	

LVSI, lympho-vascular space invasion; CA125, cancer antigen-125; PR, progesterone receptor.

**Table 2 diagnostics-12-00790-t002:** Results of univariate and multivariate logistic regression analyses in the prediction cohort.

Variable		No.	Univariate Analysis	*p*-Value	Multivariate Analysis	*p*-Value
			OR (95% CI)		OR (95% CI)	
Age (years)				0.675	-	-
	<55	149	reference			
	≥55	161	0.86 (0.42–1.76)			
Parity				0.863	-	-
	0	79	reference			
	≥1	231	1.08 (0.47–2.50)			
BMI (kg/m^2^)				0.062	-	-
	<30	229	reference			
	≥30	81	0.36 (0.12–1.05)			
CA125 (U/mL)				<0.001		<0.001
	<40	217	reference		reference	
	≥40	93	8.04 (3.57–18.13)		8.03 (3.44–18.77)	
Grade				0.001		0.009
	1	161	reference		reference	
	2/3	149	3.86 (1.68–8.85)		3.25 (1.33–7.91)	
PR (H-score)				0.001		0.002
	<162.5	172	5.40 (2.03–14.40)		5.22 (1.87–14.60)	
	≥162.5	138	reference		reference	

OR, odds ratio; CI, confidence interval; BMI, body mass index; CA125, cancer antigen-125; PR, progesterone receptor.

**Table 3 diagnostics-12-00790-t003:** Predictive performance of different variables and different combinations.

	Sensi (%)	Speci (%)	PPV (%)	NPV (%)	LR(−)	PTP(−) (%)	LR(+)	PTP(+) (%)
CA125 (cut-off 40)	72.7	75.1	25.8	95.9	-	-	-	-
Grade (1 vs. 2/3)	75.8	55.2	16.8	95.0	-	-	-	-
PR H-score (cut-off 162.5)	84.9	48	16.3	96.4	-	-	-	-
If all negative (CA125 < 40, G1, PR H-score ≥ 162.5)	97.0	21.7	-	98.4	0.14	1.5 *	-	-
If all positive (CA125 ≥ 40, G2/3, PR H-score < 162.5)	51.5	94.2	51.5	-	-	-	8.92	49.7 *

* If assumed the prevalence of LNM in endometrial cancer was 10%; PPV, positive predictive value; NPV, negative predictive value; LR(−), negative likelihood ratio; PTP(−), negative post-test probability; LR(+), positive likelihood ratio; PTP(+), positive post-test probability; CA125, cancer antigen-125; PR, progesterone receptor.

## Data Availability

Data will be available upon reasonable request.

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
