# Peer review of "Quantitative Measurement of Progesterone Receptor Immunohistochemical Expression to Predict Lymph Node Metastasis in Endometrial Cancer"

_diagnostics, 2022, doi:10.3390/diagnostics12040790_

Round 1
Reviewer 1 Report
Yu-Yang Hsiao et al. in their manuscript " Quantitative measurement of progesterone receptor immunohistochemical expression to predict lymph node metastasis in endometrial cancer. " looked at the role of PR expression and other clinicopathological factors in lymph node metastasis, and develop a prediction model in endometrial cancer.
Following a thorough analysis, the authors described what they found:
- This Taiwanese cohort study provides significant tools for monitoring and analyzing lymph node metastases in endometrial cancer patients in the Taiwan population, such as diagnostic, prognostic, which could be helpful to initiate a therapeutic approach.
- The nomogram can aid in decision-making with regards to lymphadenectomy in patients with endometrioid-type endometrial cancer.
- Clearly stated the current study's strengths and shortcomings.
This constitutes a comprehensive and broadly rational body of work that is appreciable. However, there are a number of concerns that can be resolved to improve the quality of the manuscript, listed below.
- The relevance of lines 60-64 in the introduction is not fully understood. Please put a connection for the sudden beginning of the CA125 treatment in the introduction section.
- Please provide the immunohistochemical analysis images to better understand the Table:1 data.
- To improve the data interpretation and better understanding I would suggest the authors to incorporate additional Statistical analysis. Such as supplementary figure 2 and 3 needs a trendline, including the correlation coefficient and the statistics indicating if the correlation coefficient is statistically significant. In addition to this necessary information, the equation of the trendline and the R2 to demonstrate the goodness of fit are useful.
- Please, revise the text for proper grammar use.
- The discussion portion should be reviewed with a better explanation for improved comprehension.
Reviewer 2 Report
In their work, Dr. Hsiao et al. attempted with an apparent success to create a new preoperative prediction model to identify EC patients at low risk of lymph node metastasis, and thus not eligible for routine pelvic/para-aortic lymphadenectomy. This is a commendable aim with possible implications for current clinical practice.
Overall, the topic is suitable for this Journal, the MS is well written in good English, and I have no particular concerns regarding the methodology chosen. The selection of the references is attractive.
One major shortcoming to be underlined is that the Authors chose to evaluate the immunohistochemical expression of progesterone receptor using an antibody not discerning between various progesterone receptor isoforms. A sentence of explanation about this fact should be added.
Minor issues:
Line 50, is: ‘ … control … ‘; should be: ‘ … controlled … ‘;
Legend to Figure 1: ‘consort’ should be capitalized.
